

# Recognizing two new *Hippolyte* species (Decapoda, Caridea, Hippolytidae) from the South China Sea based on integrative taxonomy

Zhi-Bin Gan[1,2] and Xin-Zheng Li[1,2,3,4]

[1] Institute of Oceanology, Chinese Academy of Sciences, Qingdao, China
[2] Center for Ocean Mega-Science, Chinese Academy of Sciences, Qingdao, China
[3] University of Chinese Academy of Sciences, Beijing, China
[4] Laboratory for Marine Biology and Biotechnology, Pilot National Laboratory for Marine Science and Technology (Qingdao), Qingdao, China

## ABSTRACT

*Hippolyte* shrimps exhibit abundant biological diversity and display great ecological significance in seaweed bed ecosystems. Dozens of *Hippolyte* specimens were collected from Hainan Island and the Xisha Islands in the South China Sea. Detailed examination indicates that some of these specimens represent new *Hippolyte* species. Based on morphological, genetic, and ecological data, *Hippolyte chacei* sp. nov. and *H. nanhaiensis* sp. nov. are described. *H. chacei* sp. nov. was collected from the *Sargassum* sp. biotope in Hainan Island and is distinguished from congeners by its unique mandible and the dactylus of the third to fifth pereiopods; this species has a basal position in the Indo-West Pacific species clade in the phylogenetic tree which is reconstructed based on 16S rRNA gene. *H. nanhaiensis* sp. nov. was collected from the biotopes of *Galaxaura* sp. or *Halimeda* sp. in the Xisha Islands, and it differs from congeners in a series of characters associated with rostrum, scaphocerite, antennular peduncle, and spines on the dactylus of the third to fifth pereiopods. Additionally, it is sister to *H. australiensis* in the phylogenetic tree. A key to identifying mature female *Hippolyte* species of the Indo-West Pacific and neighboring seas is provided.

## INTRODUCTION

Shrimps of the genus *Hippolyte* Leach, 1814 display high diversity in morphology, coloration, and ecological habits. They occur mainly in tropical and temperate oceans, although some species, such as *Hippolyte varians* Leach, 1814, are known from the Arctic Circle (*D'Udekem D'Acoz, 2007*). Most *Hippolyte* species inhabit seaweed beds, but some are obligate or facultative symbionts of other organisms, such as gorgonians and crinoids (*D'Udekem D'Acoz, 2007*; *Marin, Okuno & Chan, 2011*). The taxonomy, phylogeny, and biology of *Hippolyte* taxa have attracted considerable attention in recent

Corresponding authors
Zhi-Bin Gan,
ganzhibin2005@163.com
Xin-Zheng Li, lixzh@qdio.ac.cn

**Table 1 Information on biodiversity.**

| Sampling locality | Investigation date | Botopes |
|---|---|---|
| Hainan Island | November 20, 2014–December 01, 2014 | *Sargassum* sp. and *Zostera* sp. |
| Hainan Island | May 04, 2015–May 10, 2015 | *Sargassum* sp. and *Zostera* sp. |
| Hainan Island | April 22, 2016–April 27, 2016 | *Sargassum* sp. and *Zostera* sp. |
| Hainan Island | September 16, 2017–September 22, 2017 | *Sargassum* sp. and *Zostera* sp. |
| Hainan Island | March 20, 2018–March 28, 2018 | *Sargassum* sp. and *Zostera* sp. |
| Xisha Islands | May 12, 2015–May 19, 2015 | *Galaxaura* sp. and *Halimeda* sp. |
| Xisha Islands | July 10, 2016–July 20, 2016 | *Galaxaura* sp. and *Halimeda* sp. |

Note:
The information on biodiversity surveys.

years (*Manjón-Cabeza, Cobos & García Raso, 2011*; *Marin, Okuno & Chan, 2011*; *Terossi & Mantelatto, 2012*; *Liasko et al., 2015*; *Duarte & Flores, 2017*; *Duarte et al., 2017*; *Gan & Li, 2017a, 2017b*; *Liasko, Anastasiadou & Ntakis, 2018*; *Terossi, De Grave & Mantelatto, 2017*). Currently, a total of 35 valid species are recognized worldwide (*D'Udekem D'Acoz, 1996*, *2007*; *De Grave & Fransen, 2011*; *Marin, Okuno & Chan, 2011*; *Gan & Li, 2017a, 2017b*; *Terossi, De Grave & Mantelatto, 2017*), of which about 12 species occur in the Indo-West Pacific region. Additional unnamed and cryptic species have been reported (*Hayashi, 1986*; *D'Udekem D'Acoz, 1996*, *2007*; *Terossi, De Grave & Mantelatto, 2017*).

Due to its morphological diversity and characters overlap, and the complexity of information in published literature, taxonomic research of *Hippolyte* based on morphological characters is difficult (*D'Udekem D'Acoz, 1996*; *Gan & Li, 2017a*). The situation is particularly complicated in a complex of species referred to "*H. ventricosa* H. Milne Edwards, 1837," including: *H. acuta* (*Stimpson, 1860*), *H. australiensis* (*Stimpson, 1860*), *H. ngi* Gan & Li, 2017, *H. singaporensis* Gan & Li, 2017, *H. ventricosa* H. Milne Edwards, 1837, and *Hippolyte* sp. A from Australia, *Hippolyte* sp. B from Hawaii, *Hippolyte* sp. C from the Malay Archipelago, and *Hippolyte* sp. D from Madagascar (*D'Udekem D'Acoz, 1996*; *Gan & Li, 2017a, 2017b*). Genetic analysis (*Terossi, De Grave & Mantelatto, 2017*) has also recently detected four cryptic or pseudocryptic species referred to: *H. ventricosa* group-sp. 1 and sp. 2 from Indonesia, *H. ventricosa* group-sp. 3 from Fiji, and *H. ventricosa* group-sp. 4 from Taiwan, all with morphological features closely similar to *H. ventricosa* redescribed by *D'Udekem D'Acoz (1999)*.

During recent biodiversity surveys of Hainan Island and the Xisha Islands (2014–2018) in the South China Sea, dozens of *Hippolyte* specimens were collected. After detailed examination and multiple analyses, we described two new species of the "*H. ventricosa* H. Milne Edwards, 1837" species complex based on integrative methods.

# MATERIALS AND METHODS
## Sample collection and morphological examination

The information on biodiversity surveys are listed in Table 1. Although seaweed beds and coral reefs were sampled, species of *Hippolyte* were found only among algae and sea grass (*Sargassum* sp., *Zostera* sp., *Galaxaura* sp., and *Halimeda* sp.). All specimens were

**Table 2 Specimens information.**

| Species | Sampling locality | Voucher ID | GenBank accession numbers |
|---|---|---|---|
| *Hippolyte* cf. *ventricosa* | Hainan Island | MBM285012 | MK231003 |
| *Hippolyte* cf. *ventricosa* | Hainan Island | MBM285013 | MK231004 |
| *Hippolyte* cf. *ventricosa* | Hainan Island | MBM285014 | MK231009 |
| *Hippolyte chacei* sp. nov. | Hainan Island | MBM285015 | MK231007 |
| *Hippolyte chacei* sp. nov. | Hainan Island | MBM285016 | MK231008 |
| *Hippolyte nanhaiensis* sp. nov. | Xisha Islands | MBM285018 | MK231005 |
| *Hippolyte nanhaiensis* sp. nov. | Xisha Islands | MBM285019 | MK231006 |

Note:
Specimens collected in this study with reference to their locality, Voucher ID, and GenBank accession numbers.

collected using handheld nets while snorkeling. After being photographed, specimens were preserved in 95% ethanol. Dissection and illustrations were carried out using Nikon stereo- and compound microscopes (SMZ 1500 and AZ100). Measurements and length ratios were calculated following *D'Udekem D'Acoz (1996)*. All specimens are deposited in the Marine Biological Museum of the Chinese Academy of Sciences (MBM), in the Institute of Oceanology of Chinese Academy of Sciences, Qingdao, China.

## Molecular data and analysis

Total genomic DNA was extracted from pleopods of specimens using a QIAamp DNA Mini Kit (QIAGEN, Hilden, Germany), following manufacturer instructions. Extracted DNA was eluted in 100 µl of double-distilled $H_2O$ (dd$H_2O$). Partial sequences of 16S rRNA genes were amplified from the diluted DNA via polymerase chain reaction (PCR). Reactions were carried out in a 50-µl volume containing: 25 µl Premix Taq (TaKaRa Taq$^{TM}$ Version 2.0 plus dye; TaKaRa, Kusatsu, Japan), one µl each of forward and reverse primers (10 µM), respectively, three µl DNA template, and 20 µl dd$H_2O$. Primers 16S-AR/1472 (5′-CGCCTGTTTATCAAAAACAT-3′/5′-AGATAGAAACC AACCTGG-3′) was used (*Crandall & Fitzpatrick, 1996*). The PCR profile involved: 3 min at 94 °C for initial denaturation, 35 cycles of denaturation at 94 °C for 30 s, annealing at 52 °C for 40 s, elongation at 72 °C for 50 s, and final extension at 72 °C for 10 min. PCR products were purified using a QIAquick Gel Extraction Kit (QIAGEN, Hilden, Germany), and bidirectionally sequenced using the same primers with an ABI 3730xl Analyzer (Applied Biosystems, Foster City, CA, USA). Sequences were checked and proofread by ContigExpress 6.0 (a component of the Vector NTI Suite 6.0).

In addition to sequences obtained by PCR (Table 2; Dataset S1), we downloaded *Hippolyte* sequences from Genbank, including those previously reported cryptic or pseudocryptic taxa namely *H. ventricosa* group-sp. 1 (KX588914), *H. ventricosa* group-sp. 2 (KX588915), *H. ventricosa* group-sp. 3 (KX588915), and *H. ventricosa* group-sp. 4 (KX588915) of *Terossi, De Grave & Mantelatto (2017)*, and *H. ventricosa* group-sp. 5 (KF023090) of *De Grave et al. (2014)*.

Molecular data (Dataset S2), including 37 sequences of 16S rRNA genes, were aligned using MUSCLE 3.8 (*Edgar, 2004*). Highly divergent and poorly aligned sites were omitted

from alignment according to Gblocks 0.91b (*Castresana, 2000*). The best-fitting nucleotide base substitution model (GTR+I+G) for the alignment data was determined by Modeltest 3.7 (*Posada & Crandall, 1998*). A maximum likelihood tree was constructed using PhyML 3.1 (*Guindon & Gascuel, 2003*) with 1,000 bootstrap reiterations. A Bayesian inference tree was constructed using MrBayes 3.2 (*Huelsenbeck & Ronquist, 2001*). Markov chains were run for 2,000,000 generations, sampled every 2,000 generations; the first 25% trees were discarded as burn-in, after which remaining trees were used to construct the 50% majority-rule consensus tree and to estimate posterior probabilities. Genetic distances were calculated using the Kimura 2-parameter model in MEGA 7.0 (*Kumar, Stecher & Tamura, 2016*).

### Ecological data

The biotope (mainly the algal colony) in which a shrimp lived was recorded on capture.

The following abbreviations are used: CL, carapace length, the length from the posterior orbital margin to the posterior dorsal border of the carapace; Coll., collector (s).

The electronic version of this article in portable document format will represent a published work according to the International Commission on Zoological Nomenclature (ICZN), and hence the new names contained in the electronic version are effectively published under that Code from the electronic edition alone. This published work and the nomenclatural acts it contains have been registered in ZooBank, the online registration system for the ICZN. The ZooBank Life Science Identifiers (LSIDs) can be resolved and the associated information viewed through any standard web browser by appending the LSID to the prefix http://zoobank.org/. The LSID for this publication is urn:lsid: zoobank.org:pub:1186ACB4-410C-4061-BE93-97CE040F0702. The online version of this work is archived and available from the following digital repositories: PeerJ, PubMed Central, and CLOCKSS.

## RESULTS

### Taxonomy

**Order Decapoda Latreille, 1802**
**Family Hippolytidae Spence Bate, 1888**
**Genus *Hippolyte* Leach, 1814**
*Hippolyte chacei* sp. nov.
(Figs. 1–4 and 5A)

**Material examined.** *Holotype*: MBM285015, non-ovigerous female, 3.3 mm CL, Hongtang bay, Hainan Island, northern South China Sea, one to three m, Coll. Z B, Gan, March 25, 2018 (GenBank accession number of 16S rRNA gene: MK231007). *Paratypes*: MBM285016, one male, 2.3 mm CL, same collection data as holotype (GenBank accession number of 16S rRNA gene: MK231008); MBM285017, two non-ovigerous female, 2.7–3.0 mm CL, Houhai bay, Hainan Island, northern South China Sea, two to three m, Coll. Z B. Gan, March 22, 2018.

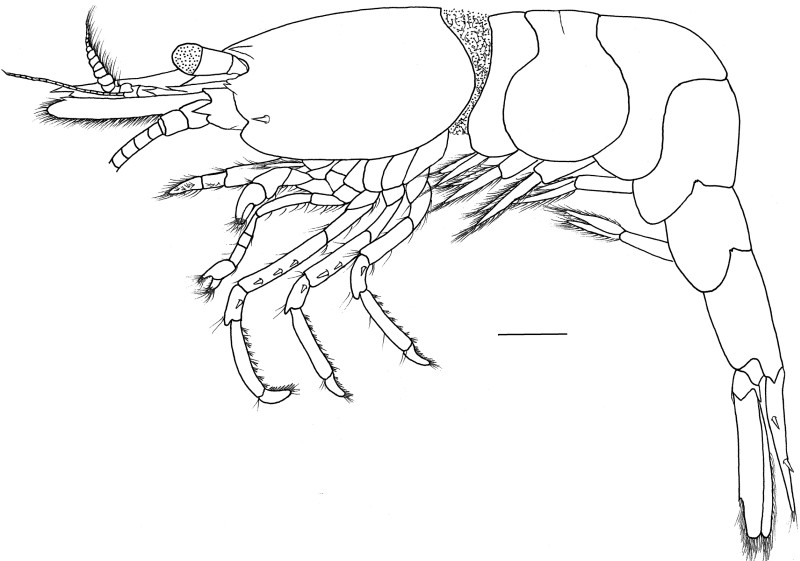

**Figure 1 *Hippolyte chacei* sp. nov. female, holotype.** *Hippolyte chacei* sp. nov. female, holotype, MBM285015, lateral view. Scale: 1.0 mm.

**Description.** Outline robust (Fig. 1). Ratio lateral length/height of carapace 1.56–1.72. Rostrum long, slightly shorter than carapace, distinctly overreaching antennular peduncle, nearly reaching to the end of scaphocerite. Rostrum without lateral carina, superior border slightly concaved, unarmed in the female specimens (Figs. 2A and 2B) and armed with only one proximal tooth in the male specimen (Fig. 2C); inferior border slightly convex, armed with four teeth in the distal half length. Carapace smooth and glabrous, location of supraorbital spine behind of the posterior orbital margin; tip of antennal spine slightly overreaching inferior orbital angle; tip of hepatic spine falling short of anterior edge of carapace. Inferior orbital angle strongly produced, knob-like (Figs. 2B and 2D). Branchiostegal margin with a distinct notch. Pterygostomian region rounded, strongly produced (Figs. 1 and 2B).

Abdominal segments smooth (Fig. 1). Third abdominal segment geniculately curved. Ratio dorsal length/height of the sixth abdominal segment 1.95–2.10. Telson (Fig. 2E) longer than the sixth abdominal segment, posterior margin rounded, armed with eight strong spines, outer spines smallest, medial two longest, without intermediate spinule or seta; dorsal surface armed with two pairs of spines situated on distal 0.31–0.35 and 0.59–0.63 telson length.

Eye (Fig. 2A) well developed, tip of cornea nearly reaching to the end of first segment of antennular peduncle when extended forward; unpigmented part of eyestalk longer than broad; cornea semispherical, distinctly shorter than unpigmented part of eyestalk.

Antennular peduncle (Fig. 2F) slightly overreaching mid-length of scaphocerite. First segment of antennular peduncle with one well developed distolateral tooth, inner ventral tooth (Fig. 2G) on 0.47–0.50 of first segment (excluding distolateral tooth). Stylocerite robust, reaching to 0.56–0.62 (distolateral tooth included), or 0.69–0.75 (distolateral tooth excluded) of first segment. Second segment of antennular peduncle

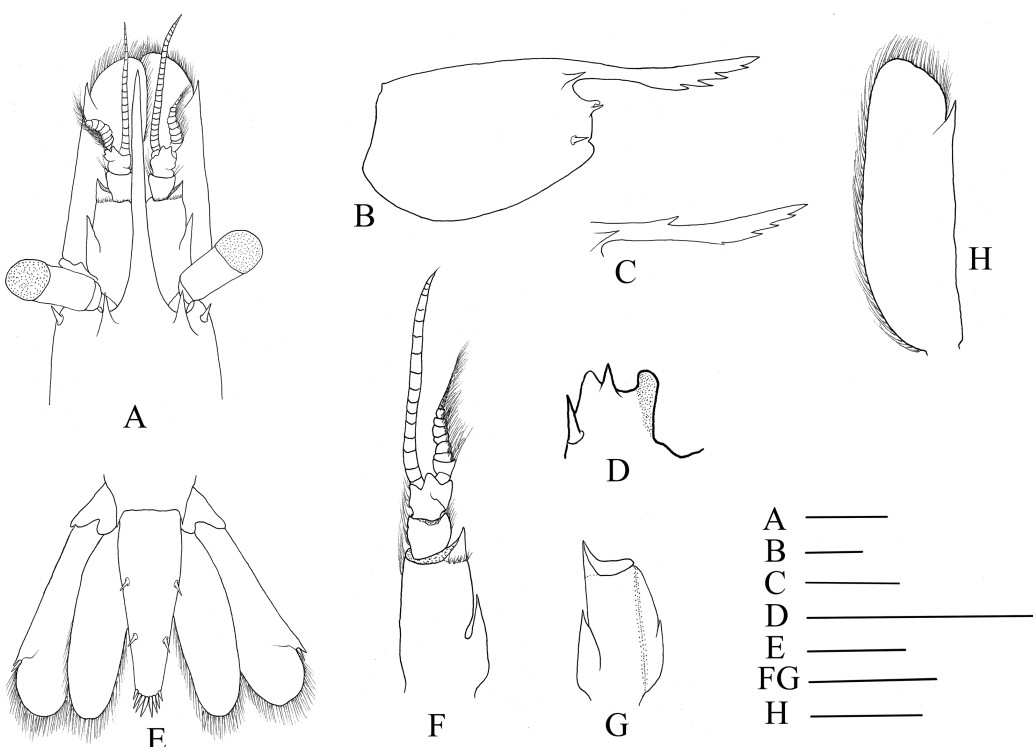

**Figure 2 Cephalon and telson of *Hippolyte chacei* sp. nov.** *Hippolyte chacei* sp. nov. (A, B, and D–H) female, holotype, MBM285015; (C) male, paratype, MBM285016. (A) Cephalon, dorsal; (B) and rostrum, lateral; (C) rostrum, lateral; (D) inferior orbital angle, dorsal; (E) telson and uropods, dorsal; (F) right antennula, dorsal; (G) first segment of right antennule, ventral; (H) right scaphocerite, dorsal. Scales: 1.0 mm.

0.81–0.86 times as long as broad in dorsal view, approximately 0.86–0.98 times as long as third segment in dorsal view. Outer antennular flagellum shorter than inner one and proximal six to eight segments thicker than distal ones. Scaphocerite (Fig. 2H) 3.06–3.18 times as long as wide, distolateral spine of scaphocerite far from reaching distal margin of blade, distolateral spine and blade separated by a distinct notch.

Mandible (Figs. 3A and 3B) without palp, incisor process with 15–17 acute teeth. Maxillula (Fig. 3C) with broad curved palp, distal margin of upper lacinia armed with 14–18 spines and scattered simple long setae. Maxilla (Fig. 3D) with short palp, scaphognathite broad and long, lateral border nearly straight; inner lacinia bilobed, distal margin furnished with row of spines and long plumose setae; proximal endite round, with long setae on distal margin. Epipod of first maxilliped (Fig. 3E) slightly bilobed; endopod broad, with distal long setae; exopod well-developed, caridean lobe broad. Second maxilliped (Fig. 3F) with well-developed exopod, flagelliform; endopod normal, dactylar segment oval, terminal margin furnished with simple and spinous setae; propodal segment with anteromedial margin round, bearing simple setae; carpus broader than long, shorter than merus; ischium and basis fused. Third maxilliped (Fig. 3G) reaching to 0.32–0.39 of scaphocerite when extended forward; exopod relatively short, only reaching to the mid-length of antepenultimate segmentof endopod; ultimate segment

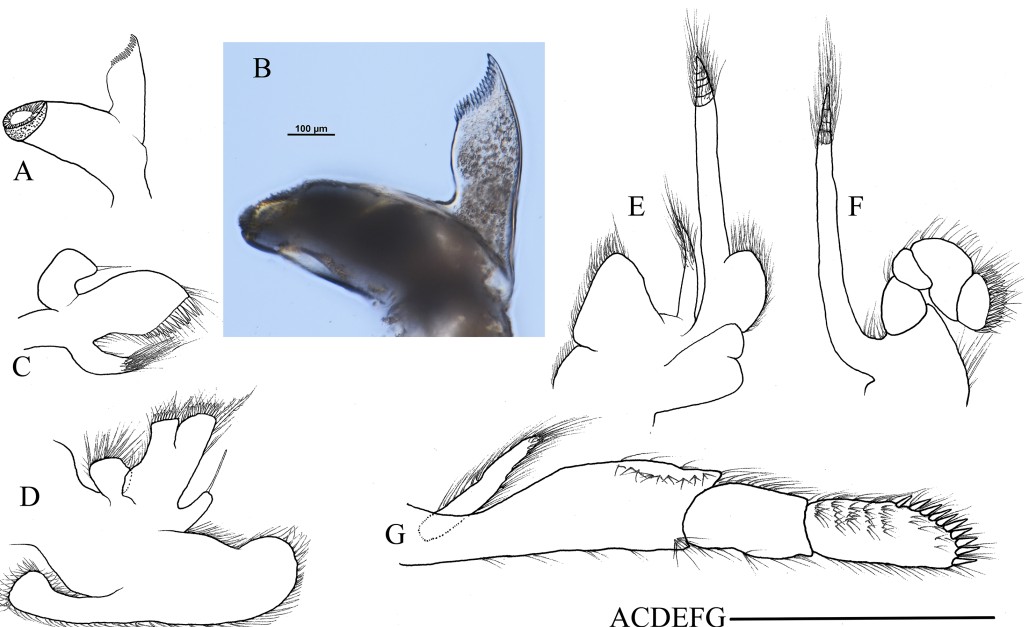

**Figure 3 Mouthparts of *Hippolyte chacei* sp. nov.** *Hippolyte chacei* sp. nov. female, holotype, MBM285015. (A and B) Left mandible; (C) left maxillule; (D) left maxilla; (E) left first maxilliped; (F) left second maxilliped; (G) left third maxilliped. Scale: 1.0 mm.

(excluding apical spine) of endopod 1.23–1.32 times as long as penultimate segment, distal half armed with seven to nine strong spines; antepenultimate segment nearly equal length to the last two segments combined.

First pereiopod (Fig. 4A) shortest among pereiopods, robust and oblique, reaching to the end of basicerite when extended forward. Ventral margin of ischium, merus, and carpus furnished with long simple setae. Terminal margin of carpus cotyloid. Cutting edges of chela non-denticulate, outer margin of fingers with long simple setae, tip of fingers armed with three acute spines, respectively (Fig. 4B).

Second pereiopod (Fig. 4C) slightly overreaching the end of third maxilliped when extended forward. Carpus with three subsegments, first subsegment 1.70–1.85 times as long as second subsegment, third subsegment slightly longer than or subequal to first subsegment; first subsegment 2.45–2.56 times as long as wide, second subsegment 1.08–1.12 times as long as wide, third subsegment 2.06–2.12 times as long as wide. Cutting edges of chela not denticulate, outer margin of fingers with long simple setae, tip of fixed finger and dactylus armed with three acute spines, respectively (Fig. 4D).

Third to fifth pereiopods long and robust. Third pereiopod (Fig. 4E) reaching to the distolateral spine of scaphocerite when extended forward; dactylus with 13–16 spines, the last two to three subdorsal spines distinctly shorter than the neighboring ones (Fig. 4F); propodus 5.56–5.62 times as long as wide, armed with six to seven pairs of spines on ventral margin; carpus 2.66–2.73 times as long as wide, armed with one proximal lateral spine; merus 5.58–5.62 times as long as wide, armed with three lateral spines. Ratio length of third pereiopod dactylus with longest apical spine/length of propodus 0.45–0.49;

2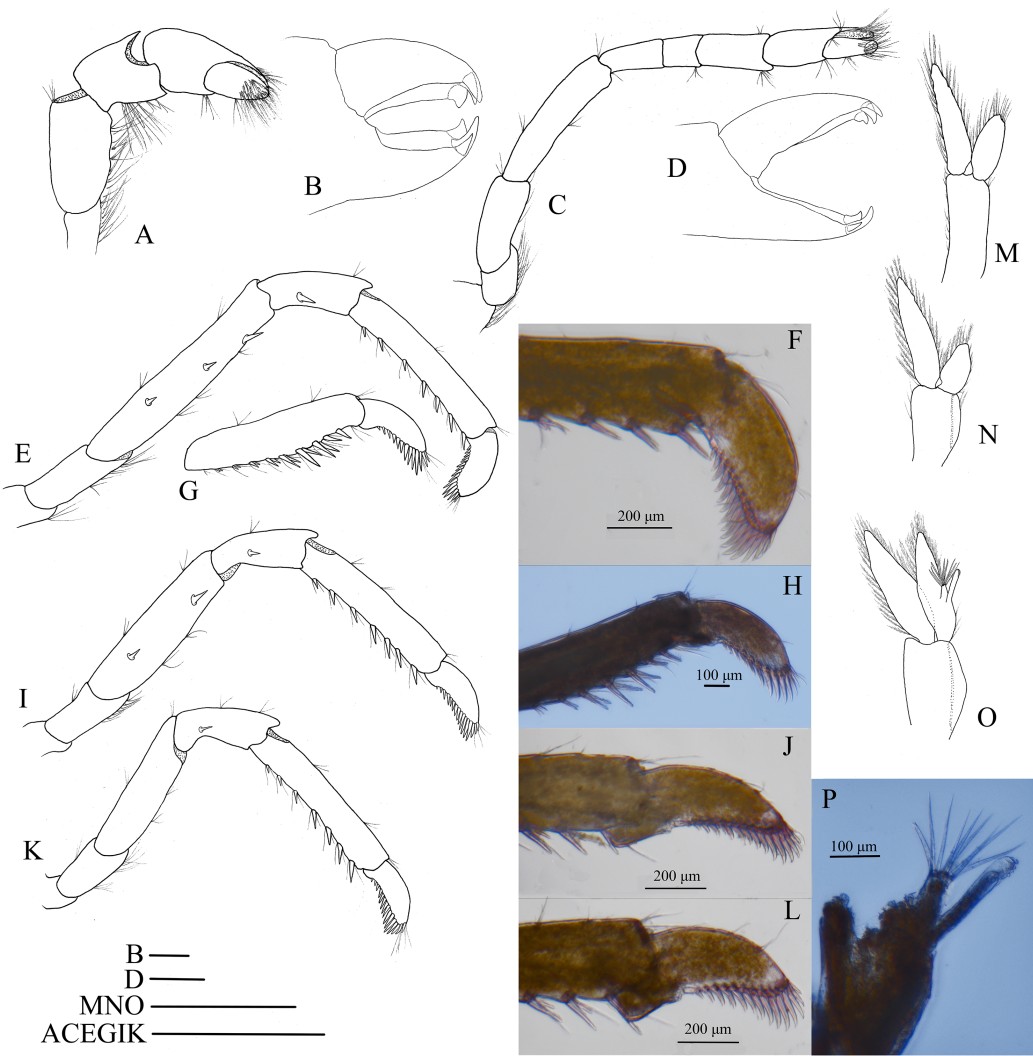

**Figure 4 Pereiopods and pleopods of *Hippolyte chacei* sp. nov.** *Hippolyte chacei* sp. nov. (A–F and I–M) female, holotype, MBM285015; (G, H, and N–P) male, paratype, MBM285016. (A) Left first pereiopod, lateral; (B) tip of the left first pereiopod, mesial (setae not shown); (C) left second pereiopod, lateral; (D) tip of the left second pereiopod, mesial (setae not shown); (E) left third pereiopod, lateral; (F) dactylus of left third pereiopod, lateral; (G and H) propodus and dactylus of left third pereiopod, lateral; (I) left fourth pereiopod, lateral; (J) dactylus of left fourth pereiopod, lateral; (K) left fifth pereiopod, lateral; (L) dactylus of left fifth pereiopod, lateral; (M and N) left first pleopod; (O) left second pleopod; (P) appendix masculina. Scales: (A, C, E, G, I, K, and M–O) one mm; (B and D) 100 µm.

ratio length of third pereiopod dactylus with longest apical spine/length of carpus 0.79–0.83; ratio length of dactylus without spines/breadth of dactylus without spines 2.61–2.69; ratio length of dactylus with longest spines/breadth of dactylus without spines 2.95–3.10; ratio length of longest spine of dactylus/breadth of dactylus without spines 0.62–0.71; ratio length of longest spine of dactylus/length of dactylus without spines 0.22–0.28. Third pereiopod (Figs. 4G and 4H) of male specimen with propodus and dactylus forming a prehensile apparatus. Fourth and fifth pereiopods (Figs. 4I–4L)

**Gan and Li (2019), *PeerJ*, DOI 10.7717/peerj.6605** 8/21

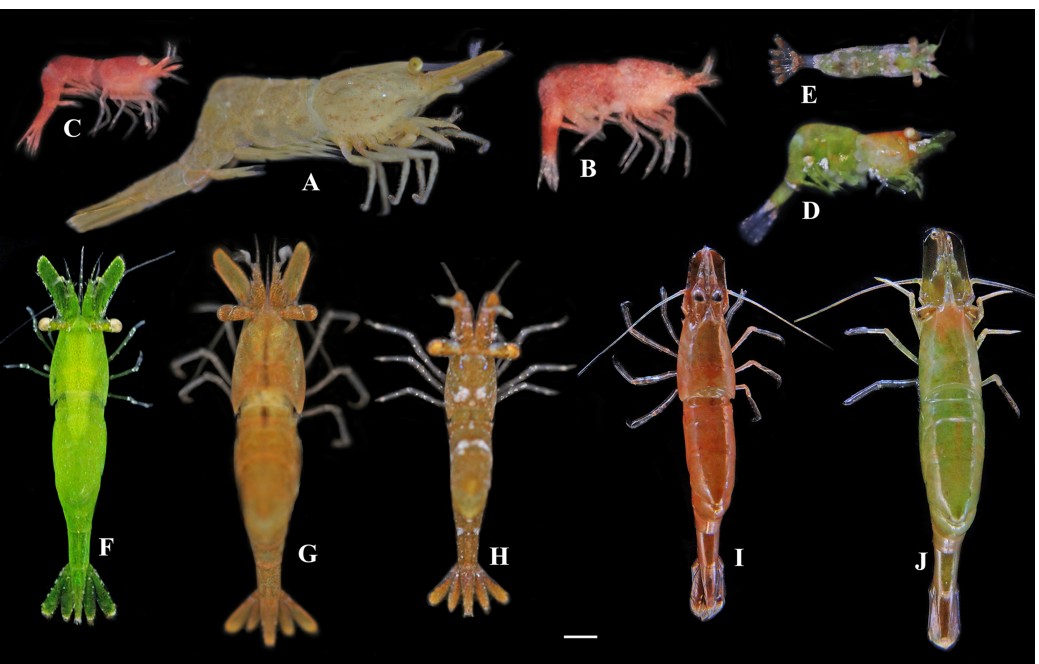

**Figure 5 Photos of *Hippolyte* spp. collected from Hainan Island and the Xisha Islands.** (A) *Hippolyte chacei* sp. nov.; (B–E) *Hippolyte nanhaiensis* sp. nov.; (F–J) *Hippolyte* cf. *ventricosa*. Scale: 1.0 mm.

similar in shape to third pereiopod of female specimen, but slightly decreasing in size. Merus of fourth pereiopod armed with two lateral spines; merus of fifth pereiopod without lateral spine.

First pleopod (Fig. 4M) of female specimen normal, endopod about 0.54–0.62 times as long as exopod. First pleopod (Fig. 4N) of male specimen with endopod about 0.41–0.46 times as long as exopod. Second pleopod (Fig. 4O) of male specimen with endopod about 0.81–0.89 times as long as exopod, appendix masculina with nine apical setae, about 0.39–0.43 times as long as appendix interna (Fig. 4P).

**Coloration.** Generally light brown over body (Fig. 5A), with few tawny stripes on carapace and faint tawny spots on abdomen.

**Biotope.** All specimens were captured among gulfweed (*Sargassum* sp.) at depths of one to three m. Numerous *Hippolyte* cf. *ventricosa* were captured simultaneously.

**Distribution.** Hongtang and Houhai bays, Hainan Island, northern South China Sea. Presumably, this species also occurs in Malayan Archipelago and Madagascar (see Discussion).

**Etymology.** The new species is named after Dr. Fenner A. Chace, Jr. in recognition of his great contribution to the crustacean taxonomy.

***Hippolyte nanhaiensis* sp. nov.**
(Figs. 6–9 and 5B–5E)

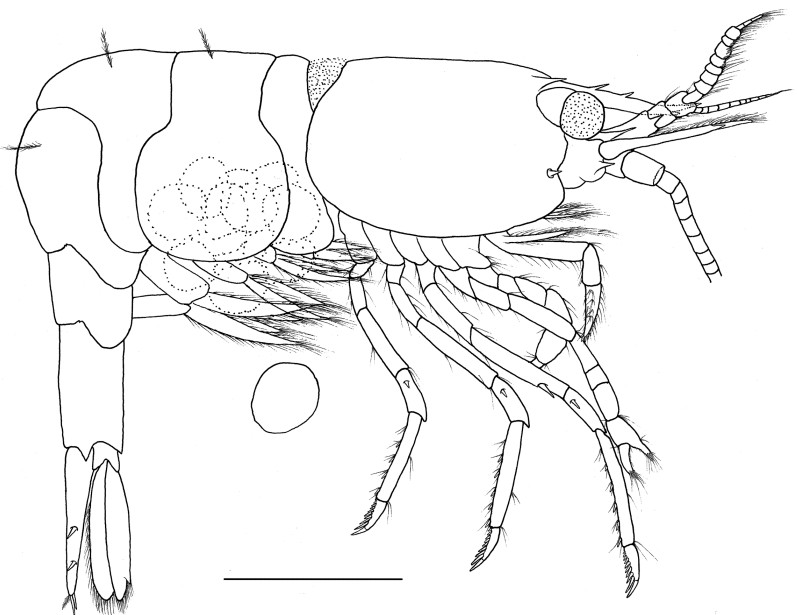

**Figure 6 *Hippolyte nanhaiensis* sp. nov. ovigerous female, holotype.** *Hippolyte nanhaiensis* sp. nov. ovigerous female, holotype, MBM285018, lateral view. Scale: 1.0 mm.

**Material examined.** *Holotype:* MBM285018, ovigerous female, 1.6 mm CL, Ganquan Island, Xisha Islands, the South China Sea, one to three m, Coll. Z B, Gan, May 15, 2015 (GenBank accession number of 16S rRNA gene: MK231005). *Paratypes:* MBM285019, one male, 1.1 mm CL, same collection data as holotype (GenBank accession number of 16S rRNA gene: MK231006); MBM189210, four ovigerous female, 1.3–1.6 mm CL, two female, 1.2–1.3 mm CL, two male, 0.9–1.1 mm CL, one juvenile 0.6 mm CL, same collection data as holotype; MBM189211, 19 ovigerous female, 1.3–1.9 mm CL, six female, 1.0–1.4 mm CL, five male, 0.8–1.1 mm CL, five juvenile 0.6–0.8 mm CL, Bei Island, Xisha Islands, the South China Sea, one to three m, Coll. Z B. Gan, May 13, 2015.

**Description.** Outline stout (Fig. 6). Ratio lateral length/height of carapace 1.49–1.58. Rostrum distinctly shorter than carapace, reaching to or slightly overreaching the end of antennular peduncle. Rostrum without lateral carina, superior border straight, armed with one or two teeth in proximal position (Figs. 7A–7D); inferior border armed with one subdistal tooth in female specimens (Fig. 7C), unarmed or only with a tiny distal notch in male specimens (Fig. 7D). Carapace smooth and glabrous. Location of supraorbital spine behind of the posterior orbital margin. Antennal spine small, slightly overreaching inferior orbital angle. Hepatic spine reaching to or slightly overreaching anterior edge of carapace. Inferior orbital angle produced, knob-like (Figs. 7B and 7C). Branchiostegal margin sinuous. Pterygostomian region rounded, strongly produced (Fig. 7C).

Abdominal segments smooth (Fig. 6), without or with few long plumose setae on tergum. Third abdominal segment geniculately curved. Ratio dorsal length/height of the sixth abdominal segment 1.91–2.08. Telson (Fig. 7E) longer than sixth abdominal segment, posterior margin rounded, armed with eight strong spines, outer spines smallest,

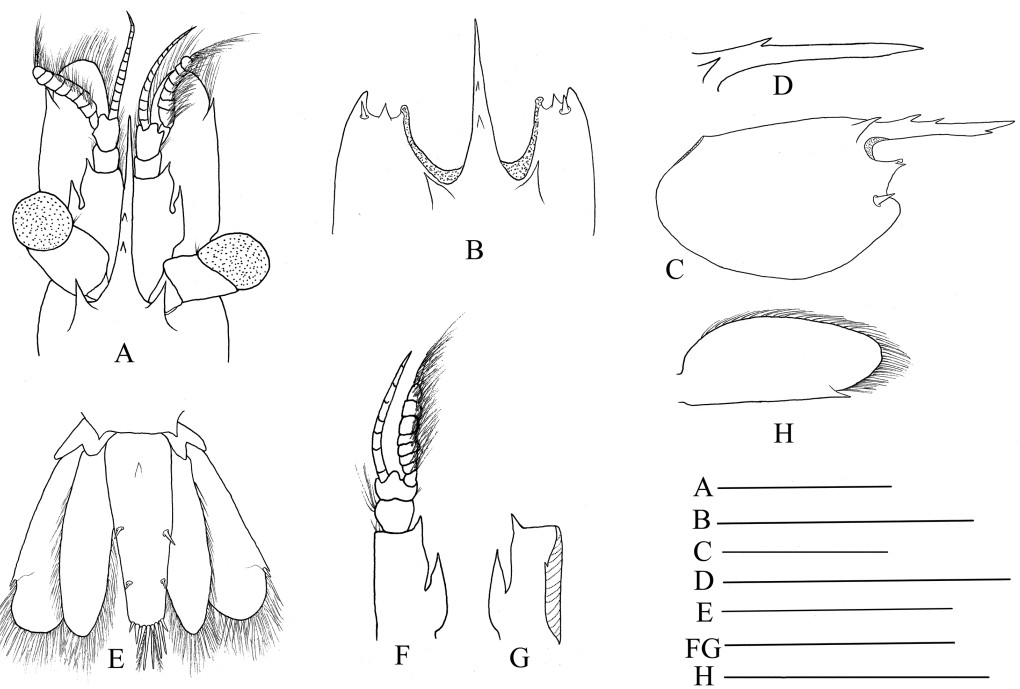

**Figure 7 Cephalon and telson of *Hippolyte nanhaiensis* sp. nov.** *Hippolyte nanhaiensis* sp. nov. (A–C and E–H) ovigerous female, holotype, MBM285018; (D) male, paratype, MBM285019. (A) Cephalon, dorsal; (B) anterior of rostrum, dorsal; (C) carapace and rostrum, lateral; (D) rostrum, lateral; (E) telson and uropods, dorsal; (F) right antennula, dorsal; (G) first segment of right antennule, ventral; (H) right scaphocerite, dorsal. Scales: 1.0 mm.

medial two longest, without or with two intermediate long plumose setae; dorsal surface armed with two pairs of spines situated on distal 0.21–0.26 and 0.43–0.49 telson length.

Eye (Fig. 7A) well developed, tip of cornea falling short of the first segment of antennular peduncle when extended forward; unpigmented part of eyestalk slightly longer than broad; cornea semispherical, slightly shorter than unpigmented part of eyestalk.

Antennular peduncle (Fig. 7F) distinctly overreaching mid-length of scaphocerite. First segment of antennular peduncle with one distolateral tooth; inner ventral tooth (Fig. 7G) on 0.59–0.66 of first segment (excluding distolateral tooth), small. Stylocerite robust, reaching to 0.86–0.92 (distolateral tooth included), or 0.76–0.81 (distolateral tooth excluded) of first segment. Second segment of antennular peduncle 0.88–0.96 times as long as broad in dorsal view, 0.83–0.95 times as long as third segment. Outer antennular flagellum shorter than inner one and proximal seven to nine segments thicker than distal ones. Scaphocerite (Fig. 7H) 2.19–2.38 times as long as wide, distolateral spine of scaphocerite far from reaching distal margin of blade, distolateral spine and blade separated by a notch.

Mouthparts with morphology typical for the genus *Hippolyte*. Mandible (Fig. 8A) without palp, incisor process with five acute teeth. Maxillula (Fig. 8B) with curved palp, distal margin of upper lacinia armed with 8–10 spines and two long plumose setae. Maxilla (Fig. 8C) with short palp; scaphognathite broad in upper half and narrow in lower half, lateral border nearly straight; inner lacinia bilobed, distal margin furnished with

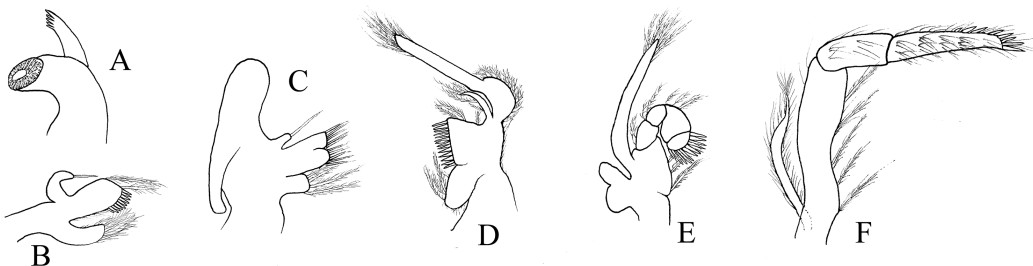

**Figure 8 Mouthparts of *Hippolyte nanhaiensis* sp. nov.** *Hippolyte nanhaiensis* sp. nov. ovigerous female, holotype, MBM285018. (A) Left mandible; (B) left maxillule; (C) left maxilla; (D) left first maxilliped; (E) left second maxilliped; (F) left third maxilliped. Scale: 1.0 mm.

spinous setae; proximal endite round, with long plumose setae on distal margin. Endopod of first maxilliped (Fig. 8D) slender, with long plumose setae; exopod with feeble caridean lobe on base. Second maxilliped (Fig. 8E) with well-developed exopod, flagelliform; endopod normal, dactylar segment arched, terminal margin armed with row of long spines; propodal segment bearing few long plumose setae; carpus longer than broad, shorter than merus. Third maxilliped (Fig. 8F) reaching to mid-length of scaphocerite when extended forward; exopod reaching to 0.72–0.79 of antepenultimate segment; ultimate segment (excluding apical spine) of endopod 1.61–1.78 times as long as penultimate segment, distal half armed with six to nine strong spines; antepenultimate segment slightly shorter than the last two segments combined.

First pereiopod (Fig. 9A) shortest among pereiopods, oblique, nearly reaching to mid-length of the scaphocerite when extended forward. Ventral margin of basis, ischium, and merus furnished with long plumose setae. Terminal margin of carpus cotyloid. Cutting edges of chela non-denticulate, outer margin of fingers with long simple setae, tip of fixed finger with three acute spines, tip of dactylus with four acute spines (Fig. 9B).

Second pereiopod (Fig. 9C) slightly overreaching the distolateral spine of scaphocerite when extended forward. Carpus with three subsegments, first subsegment 2.13–2.26 times as long as second subsegment, third subsegment slightly shorter than first subsegment; first subsegment 2.65–2.76 times as long as wide, second subsegment 1.08–1.16 times as long as wide, third subsegment 1.76–1.83 times as long as wide. Cutting edges of chela not denticulate, outer margin of fingers with long simple setae, tip of fixed finger with three acute spines, tip of dactylus with four acute spines (Fig. 9D).

Third to fifth pereiopods long and robust. Third pereiopod (Fig. 9E) reaching beyond terminal blade of scaphocerite by dactylus when extended forward; dactylus with 8–10 spines, all spines in ventral and apical position (none in dorsal or subdorsal position), with two apical spines larger than others (the ultimate one longer but thinner than the penultimate one) (Fig. 9F); propodus 6.98–7.12 times as long as wide, armed with four to six pairs of spines on ventral margin; carpus 2.96–3.14 times as long as wide, armed with one proximal lateral spine; merus 6.45–6.63 times as long as wide, armed with two lateral spines. Ratio length of third pereiopod dactylus with longest apical spine/length of propodus 0.42–0.46; ratio length of third pereiopod dactylus with longest apical

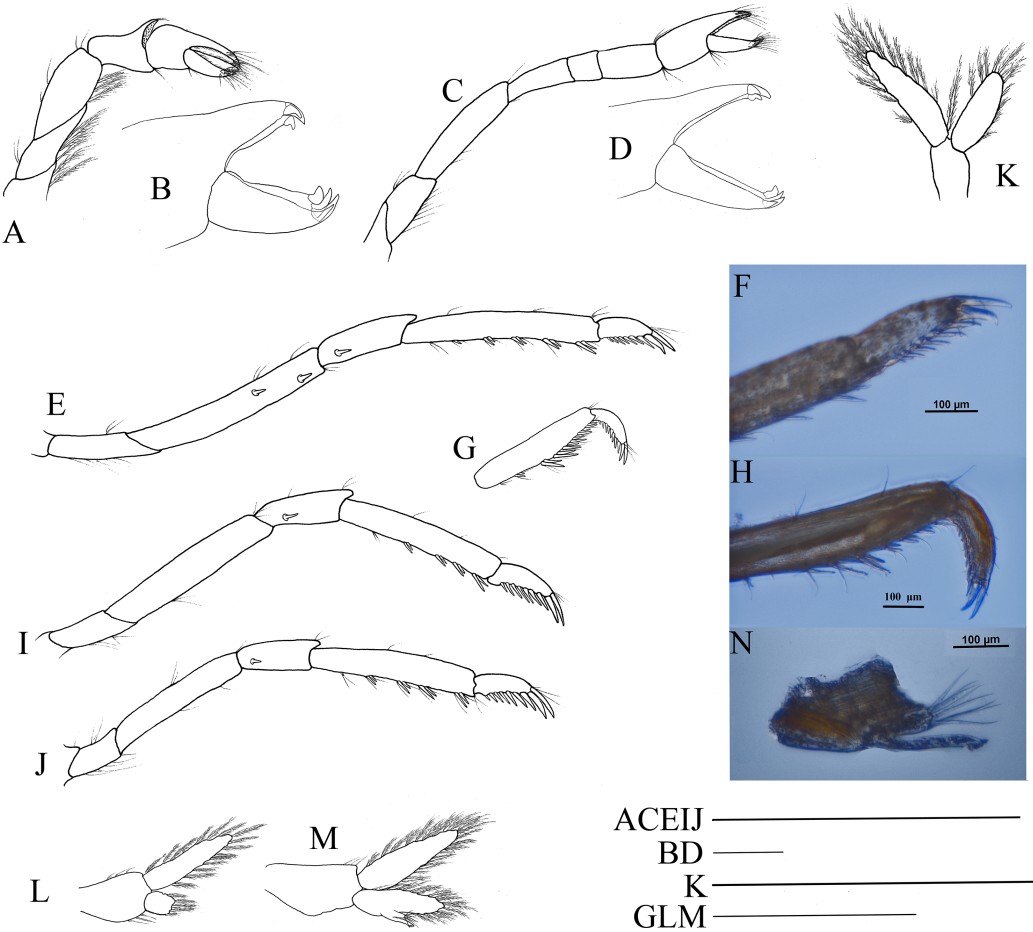

**Figure 9 Pereiopod and pleopods of *Hippolyte nanhaiensis* sp. nov.** *Hippolyte nanhaiensis* sp. nov. (A–F and I–K) female, holotype, MBM285018; (G, H, and L–N) male, paratype, MBM285019. (A) Right first pereiopod, lateral; (B) tip of the left right pereiopod, mesial (setae not shown); (C) right second pereiopod, lateral; (D) tip of the right second pereiopod, mesial (setae not shown); (E) right third pereiopod, lateral; (F) dactylus of right third pereiopod, lateral; (G and H) propodus and dactylus of right third pereiopod, lateral; (I) right fourth pereiopod, lateral; (J) right fifth pereiopod, lateral; (K and L) left first pleopod; (M) right second pleopod; (N) appendix masculina. Scales: (A, C, E, G, and I–M) one mm; (B and D) 100 μm.               

spine/length of carpus 0.86–0.92; ratio length of dactylus without spines/breadth of dactylus without spines 2.86–2.93; ratio length of dactylus with longest spines/breadth of dactylus without spines 4.35–4.43; ratio length of longest spine of dactylus/breadth of dactylus without spines 1.50–1.55; ratio length of longest spine of dactylus/length of dactylus without spines 0.53–0.58. Third pereiopod of male specimen with propodus and dactylus forming a prehensile apparatus (Figs. 9G and 9H). Fourth and fifth pereiopods (Figs. 9I and 9J) similar in shape to third pereiopod of female specimen, but slightly decreasing in size. Merus of fourth pereiopod armed with 0–1 lateral spine, merus of fifth pereiopod without lateral spine.

First pleopod (Fig. 9K) of female specimen normal, endopod about 0.72–0.78 times as long as exopod. First pleopod (Fig. 9L) of male specimen with endopod about

0.25–0.29 times as long as exopod. Second pleopod (Fig. 9M) of male specimen with endopod about 0.79–0.86 times as long as exopod; appendix masculina with eight apical setae, about 0.41–0.47 times as long as appendix interna (Fig. 9N).

**Coloration and Biotopes.** Specimens collected from different biotopes manifested different body colors. Specimens (Figs. 5B and 5C) captured among *Galaxaura* sp. were generally pink over body, with numerous white spots; specimens (Figs. 5D and 5E) captured among *Halimeda* sp. were generally light green over body, with white or pink stains on the carapace, abdomen, and telson. All specimens were captured from one to three m depth.

**Distribution.** Xisha Islands, South China Sea. Presumably, this species also occurs in Taiwan Island (see Discussion).

**Etymology.** "Nanhai" means the South China Sea; the new species is named after its type locality.

*Hippolyte* cf. *ventricosa* H. Milne Edwards, 1837
(Figs. 5F–5J)

**Material examined.** MBM189209, ovigerous female, 4.9 mm CL, Houhai bay, Hainan Island, northern South China Sea, one to three m, Coll. Z B, Gan, November 22, 2014 (GenBank accession number of 16S rRNA gene: MK231003). MBM189208, 19 male, 1.6–2.8 mm CL, 25 female, 1.8–4.7 mm CL, 36 ovigerous female, 2.6–5.0 mm CL, 11 juvenile 0.6–1.1 mm CL, same collection data as MBM189209; MBM189207, one ovigerous female, 3.8 mm CL, Hongtang bay, Hainan Island, northern South China Sea, one to two m, Coll. Z B. Gan, March 22, 2018 (GenBank accession number of 16S rRNA gene: MK231004); MBM189208, three male, 1.3–2.6 mm CL, three female, 1.5–3.9 mm CL, six ovigerous female, 2.5–4.7 mm CL, same collection data as MBM189207; MBM189206, one female, 3.5 mm CL, Luhuitou bay, Hainan Island, northern South China Sea, one to two m, Coll. Z B. Gan, May 8, 2015 (GenBank accession number of 16S rRNA gene: MK231009); MBM189205, two male, 2.0–2.6 mm CL, one female, 3.2 mm CL, four ovigerous female, 2.8–3.9 mm CL, same collection data as MBM189206; MBM189204, five male, 1.4–2.8 mm CL, seven female, 1.4–4.0 mm CL, nine ovigerous female, 2.2–4.5 mm CL, Dadonghai bay, Hainan Island, northern South China Sea, two to three m, Coll. Z B. Gan, April 25, 2016; MBM189203, four male, 1.1–2.9 mm CL, nine female, 1.3–3.8 mm CL, 15 ovigerous female, 2.3–4.4 mm CL, two juvenile 0.6–0.9 mm CL, Yalong bay, Hainan Island, northern South China Sea, one to three m, Coll. Z B. Gan, September 18, 2017.

**Remarks.** These specimens had the following features: (1) first article of the antennular peduncle with one distolateral tooth, and fifth pleonite no dorsolateral tooth; (2) carapace length of mature females among 1.8–3.3 mm, and total length among 13–24 mm; (3) rostrum distinctly overreaching the end of the antennular peduncle but falling short of scaphocerite apex, superior border with one to two teeth and inferior border with

one to five teeth; (4) incisor process of mandible with five to six teeth; (5) scaphocerite 2.79–3.38 times as long as wide; (6) dactyli of the third to fifth pereiopods with two large apical spines, but the longest apical spine never exceeding the half-length of dactyli properly, the ratio of the longest spine of dactylus/length of dactylus without spines among 0.33–0.41; (7) specimens displaying various colorations (Figs. 5F–5J).

These features differ from those described for *H. acuta*, *H. australiensis*, *H. ngi*, *H. singaporensis*, and *H. nanhaiensis* sp. nov., but are similar to the morphological characters of *H. ventricosa* (referring to the redescription of *D'Udekem D'Acoz, 1999*). More than four cryptic or pseudocryptic species were recently detected using molecular markers, which were also morphologically very similar to *H. ventricosa* (*De Grave et al., 2014*; *Terossi, De Grave & Mantelatto, 2017*). Therefore, it is not clear which specimens represent the true *H. ventricosa*; 16S rRNA or other genetic data derived from the *H. ventricosa* topotype are expected to resolve this issue.

**Coloration and Biotopes.** Specimens captured among *Thalassia* sp. were generally bright green over body (Fig. 5F), or green over body with pink or brown stains on carapace, abdomen, and telson (Fig. 5J); specimens (Figs. 5G–5I) captured among *Sargassum* sp. are generally sandy brown or reddish brown over body, with or without white stains on carapace, abdomen, and telson. All specimens were captured at depths of one to three m.

## DISCUSSION

*Hippolyte chacei* sp. nov. is distinguished from all other valid *Hippolyte* species by the unique dactylus of the third to fifth pereiopods. This type of dactylus has previously reported only for specimens attributed to *H. ventricosa*, such as those reported from the Malayan Archipelago (*Holthuis, 1947*), Madagascar (*Ledoyer, 1970*), and Hawai (*Hayashi, 1981*), which *D'Udekem D'Acoz (1996)* considered represented undescribed species. Our work, based on molecular data, confirms this suspicion. In the 16S rRNA phylogenetic tree (Fig. 10), *H. chacei* sp. nov. (two specimens) form an isolated branch clustered in the subbasal position of the Indo-West Pacific clade (*Terossi, De Grave & Mantelatto, 2017*). Additionally, the average genetic divergence between *H. chacei* sp. nov. and other *Hippolyte* species is 20.8%, which is slightly greater than the average interspecific genetic divergence of 20.5% (calculated from the 30 *Hippolyte* species in our study).

Specimens attributed to *H. ventricosa* from the Malayan Archipelago and Madagascar by *Holthuis (1947)* and *Ledoyer (1970)*, respectively, are very similar to *H. chacei* sp. nov. in morphology. We speculate that they are conspecific, but this speculation requires a detailed examination of their specimens. *Hayashi (1981)* stated that the mouthparts of Hawaiian specimens were similar to those of *H. edmondsoni* and *H. jarvisensis*, of which distinctly differ from those of *H. chacei* sp. nov.; moreover, difference is also apparent in the position of hepatic spine. Those specimens reported by *Hayashi (1981)* may represent a different species from *H. chacei* sp. nov.

Morphologically, *H. nanhaiensis* sp. nov. is similar to *H. acuta*, *H. australiensis*, *H. ngi*, *H. singaporensis*, and *H. ventricosa* (referring to the redescription of *D'Udekem D'Acoz, 1999*).

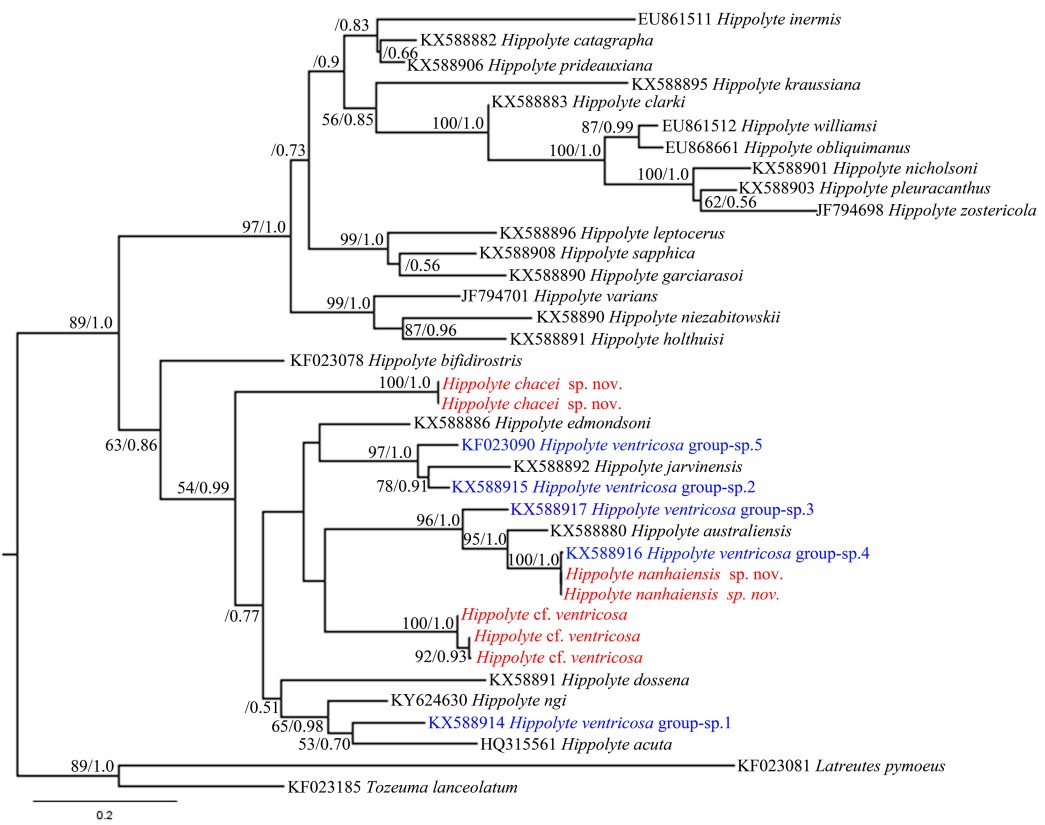

**Figure 10 16S rRNA phylogenetic tree.** Maximum likelihood tree based on 16S rRNA sequence data. Numbers at nodes represent posterior probabilities/bootstrap values (BI/ML), numbers less than 50 or 0.50 are not shown.

They all have the first article of the antennular peduncle with one distolateral tooth, fifth pleonite without dorsolateral teeth, and third to fifth pereiopods with normal dactyli. *H. nanhaiensis* sp. nov. differs from *H. acuta*, *H. australiensis*, and *H. ngi* by its shorter rostrum (reaching to or only slightly overreaching the end of the antennular peduncle vs. distinctly overreaching the end of the antennular peduncle). *H. acuta* is further distinguished from *H. nanhaiensis* sp. nov. by its particularly long eyestalk (*Stimpson, 1860*; *Hayashi & Miyake, 1968*; *Yanagawa & Watanabe, 1988*; *D'Udekem D'Acoz, 1996*). *H. australiensis* is further distinguished from *H. nanhaiensis* sp. nov. by its rostrum, which has a sharp lateral carina, and also by the dactylus of the third to fifth pereiopods, which have four large apical spines (*D'Udekem D'Acoz, 2001*). *H. ngi* differs from *H. nanhaiensis* sp. nov. by its hepatic, which overreaches the anterior edge of carapace by distal half length, and also by the dactylus of the third to fifth pereiopods, which have three large apical spines (*Gan & Li, 2017b*).

According to *D'Udekem D'Acoz (1999)*, *H. ventricosa* also has two large apical spines on the dactylus of the third to fifth pereiopods, although the apical spines of *H. nanhaiensis* sp. nov. are proportionally longer. The ratio of the longest spine of the dactylus/length of the dactylus without spines is 0.53–0.58 in *H. nanhaiensis* sp. nov., but it is only 0.35 in *H. ventricosa*. The rostrum of *H. ventricosa* distinctly overreaches the end of the antennular

peduncle, but it only reaches to or slightly overreaches the end of the antennular peduncle in *H. nanhaiensis* sp. nov. The scaphocerite of *H. ventricosa* is 3.10 times as long as wide, but it is 2.19–2.38 times as long as wide in *H. nanhaiensis* sp. nov. The total length of the *H. ventricosa* syntypes reaches 17 mm (*D'Udekem D'Acoz, 1999*), nearly two times longer than the largest *H. nanhaiensis* sp. nov. Furthermore, the two species inhabit different ecological niches. *H. ventricosa* lives among *Zostera* sp. or *Padina* sp., and may also be found among *Sargassum* sp., but *H. nanhaiensis* sp. nov. was found only among *Galaxaura* sp. or *Halimeda* sp., and no other congeners were found in these biotopes.

In the 16S rRNA phylogenetic tree (Fig. 10), *H. nanhaiensis* sp. nov. (two specimens) form a clade with *H. ventricosa* group-sp. 4 (*Terossi, De Grave & Mantelatto, 2017*), with this clade being a sister to *H. australiensis*. The average genetic divergence between *H. nanhaiensis* sp. nov. and other *Hippolyte* species is 22.5%, which is greater than the average interspecific genetic divergence (20.5%). 16S rRNA sequence alignment reveals *H. nanhaiensis* sp. nov. to be identical to, or to has a single nucleotide base difference from specimen of *H. ventricosa* group-sp. 4 (KX588916). Therefore, the specimen attributed to *H. ventricosa* group-sp. 4 and *H. nanhaiensis* sp. nov. might ultimately prove to be conspecific.

## CONCLUSIONS

As noted by *D'Udekem D'Acoz (1999* and *2001)*, the systematics of Indo-West Pacific *Hippolyte* is extremely complicated, even though this region is considered the origin center of the genus (*Terossi, De Grave & Mantelatto, 2017*). Much of this taxonomic confusion stems from a lack of knowledge of several species, such as *H. proteus*, *H. kraussiana*, and *H. acuta*, and the plasticity in morphological characters of deemed taxonomic importance. Our study demonstrate the length of the rostrum relative to the antennular peduncle, the ratio of width to height of the scaphocerite, the position of the hepatic spine, and the features of the dactylus of the third to fifth pereiopods to be taxonomic value.
A preliminary key for the indentification of mature female of the genus *Hippolyte* occurring in the Indo-West Pacific and neighboring seas is provided. This key only contains valid species listed in WoRMS (http://www.marinespecies.org); the cryptic or pseudocryptic *H. ventricosa* species are temporarily pooled as "*H. ventricosa*" *sensu lato*.

### Key to mature female of *Hippolyte* for the Indo-West Pacific and neighboring seas

1a-First segment of antennular peduncle without distolateral tooth ..................... 2

1b-First segment of antennular peduncle with one distolateral tooth ................... 3

2a-Merus of third pereiopod with no more than one lateral spine, scaphocerite about 2.8 times as long as wide ................................................... *H. proteus*

2b-Merus of third pereiopod with 3–5 lateral spines, scaphocerite about 3.5 times as long as wide ................................................................ *H. kraussiana*

3a-Dactyli of third to fifth pereiopods slender, simple, with elongate curved unguis, without ventral spines, mainly associated with Alcyonacean corals ..................... 4

3b-Dactyli of third to fifth pereiopods with obvious ventral or subdorsal spines, mainly inhabited among seaweeds ......................................................... 5

4a-Carapace with dorsal surface strongly gibbous, fingers of first pereiopod about half as long as palm ...................................................... *H. dossena*

4b-Carapace with dorsal surface flat, non-gibbous, fingers of first pereiopod subequal to palm ...................................................... *H. commensalis*

5a-Rostrum not overreaching the end of antennular penduncle ........................ 6

5b-Rostrum distinctly overreaching the end of antennular penduncle ................. 10

6a-Rostrum less than half length of carapace, reaching to the end of first segment of antennular penduncle at most .............................................*H. edmondsoni*

6b-Rostrum longer than the half length of carapace, reaching to the end of antennular penduncle ............................................................................... 7

7a-Rostrum without dorsal tooth, base of hepatic spine nearly situating at anterior edge of carapace ...................................................................... *H. singaporensis*

7b-Rostrum with 1-2 dorsal teeth, base of hepatic spine situating at posterior to the anterior edge of carapace ......................................................... 8

8a-Distal spine of dactylus of third pereiopod longer than the half length of dactylus proper (excluding spines) .......................................................... 9

8b-Distal spine of dactylus of third pereiopod shorter than the half length of dactylus proper (excluding spines) ................................................. *H. jarvinensis*

9a-Rostrum with postrostral spine, situating at just above the orbit ..........*H.caradina*

9b-Rostrum without postrostral spine, all dorsal spines situating at prior to the orbit *H. nanhaiensis* sp. nov.

10a-Incisor process of mandible with no more than 8 acute teeth .................... 11

10b-Incisor process of mandible with 15–17 acute teeth, dactyli of third to fifth pereiopods with 2–3 subdorsal spines ......................................... *H. chacei* sp. nov.

11a-Unpigmented part of eyestalk 3 times as long as cornea ................... *H. acuta*

11b-Unpigmented part of eyestalk no more than 3 times as long as cornea ........... 12

12a-Rostrum without dorsal spine ................................................... 13

12b-Rostrum with dorsal spine ..................................................... 14

13a-Apex of the rostrum trifid ......................................... *H. multicolorata*

13b-Apex of rostrum simple ............................................. *H. australiensis*

14a-Apex of the Rostrum bifid .......................................... *H. bifidirostris*

14b-Apex of rostrum simple ........................................................ 15

15a-Dactyli of third to fifth pereiopods with three long terminal teeth, distal half of hepatic spine overreaching anterior edge of carapace ................................... *H. ngi*

15b-Dactyli of third to fifth pereiopods with two terminal teeth, hepatic spine slightly overreaching anterior edge of carapace........................... *H. ventricosa sensu lato*

## ACKNOWLEDGEMENTS

We are extremely grateful to Dr. Xinming Liu (Guangxi Academy of Oceanography) and Dr. Dong Dong (Institute of Oceanology, Chinese Academy of Sciences) for their kind help with photographing the specimens, and sincere thanks are extended to associate professor Yuanchao Li (Hainan Academy of Ocean and Fisheries Sciences) for his great help in the sample collection in the Xisha Islands.

### Funding

This work was supported by the Public Science and Technology Research Funds Projects of Ocean (No. 201505004) and the National Natural Science Foundation of China (Nos. 41506171 and 30370186). The funders had no role in study design, data collection and analysis, decision to publish, or preparation of the manuscript.

### Grant Disclosures

The following grant information was disclosed by the authors:
Public Science and Technology Research Funds Projects of Ocean: 201505004.
National Natural Science Foundation of China: 41506171 and 30370186.

### Competing Interests

The authors declare that they have no competing interests.

### Author Contributions

- Zhi-Bin Gan conceived and designed the experiments, performed the experiments, analyzed the data, contributed reagents/materials/analysis tools, prepared figures and/or tables, authored or reviewed drafts of the paper, approved the final draft.
- Xin-Zheng Li conceived and designed the experiments, authored or reviewed drafts of the paper.

### DNA Deposition

The following information was supplied regarding the deposition of DNA sequences:
The 16S rRNA sequences described here are accessible via GenBank accession numbers MK231003–MK231009.

### Data Availability

The raw data for constructing phylogenetic tree is available in the Supplemental Files.

### New Species Registration

The following information was supplied regarding the registration of a newly described species:
Publication LSID: urn:lsid:zoobank.org:pub:1186ACB4-410C-4061-BE93-97CE040F0702

Hippolyte chacei Gan & Li LSID: urn:lsid:zoobank.org:act:2A219926-8CEC-4106-930A-8CF7EB14417E

Hippolyte nanhaiensis Gan & Li LSID: urn:lsid:zoobank.org:act:A0FDA677-E061-448A-A323-DEFD5AF23C72

## Supplemental Information

Supplemental information for this article can be found online at http://dx.doi.org/10.7717/peerj.6605#supplemental-information.

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
