# Peer review of "Recognizing two new Hippolyte species (Decapoda, Caridea, Hippolytidae) from the South China Sea based on integrative taxonomy"

_PeerJ, doi:10.7717/peerj.6605_

## Round 0.1 · original submission · Major Revisions

I read the MS and agree that it presents novel information regarding Hippolyte species. I'm especially happy to see the description of two new species in an integrative approach, using both molecular and morphological methods. However, I agree with the reviewers that major revisions are needed. Also, the new version should be given a thorough revision of the English language by a professional.

Additionally, I recommend to revise the MS taking into account all the suggestions made, especially the need to have a table with morphological characters or a key in order to summarize knowledge so far and to serve as help for other researchers working on Hippolyte ventricosa group and the improvement of the quality of the photos. Please, when answering provide a letter explaining step by step all the changes made. I look forward reading the new version.

·

Basic reporting

(1) Clear, unambiguous, professional English
language used throughout.

NO. With the exception of the description, the English is quite poor. This is the major problem with this paper. See General comments for the authors.

(2) Intro & background to show context.
Literature well referenced & relevant.

YES

(3) Structure conforms to PeerJ standards,
discipline norm, or improved for clarity.

YES

(4) Figures are relevant, high quality, well
labelled & described.

YES

(5) Raw data supplied (as required by PeerJ policy).

YES

Experimental design

(1) Original primary research within Scope of
the journal.

I think so.

(2) Research question well defined, relevant
& meaningful. It is stated how the
research fills an identified knowledge gap.

YES

(3) Rigorous investigation performed to a
high technical & ethical standard.

YES

(4) Methods described with sufficient detail &
information to replicate.

YES

Validity of the findings

No problems found. This is a good classical and modern taxonomic paper, where everything has been done properly.

Additional comments

(1) general scientific review

This paper is a valuable and significant contribution to the taxonomic knowledge of Indo-Pacific Hippolyte species. The authors use a modern integrative approach and the scientific quality of the paper is good. The interpretations and analyses of the paper are usually correct. The structure of the paper is adequate and the style is usually clear. In other words, it is a good modern classical taxonomical paper (not highly original but scientifically well made).
It certainly deserves publication (after language improvement). Whether PeerJ is the appropriate journal or not should be double-checked by the editor. However I think it is.

I would add a key or a character table for Indo-pacific Hippolyte of the "group ventricosa" (to which your two new species do belong). That key or table do not have to be too elaborate (as more species will probably be discovered). For example, problematic taxa like the cryptic species currently confused under the name Hippolyte ventricosa could be pooled together as "Hippolyte ventricosa sensu lato". At this stage, even a very basic table/key would be of considerable help for other carcinologists working on that difficult shrimp group.

The line drawings are correctly executed.

On the plates, the photographs are sometimes weakly contrasted (a problem often found with raw photographs of crustacean appendages). Usual imaging softwares have functions that can be used for increasing the contrasts of such structures. With Adobe Photoshop, the most useful function is Shadow/Highlight. I imagine that GIMP has an equivalent function but I am not familiar with that software. I would recommend to the authors to test such functions and, if possible, to submit photographs with increased contrasts in the second version of the manuscript. This is not absolutely necessary but this would be highly preferable.

(If the authors intend to publish photographs in further papers, I would recommend them to test stacking photography, which considerably improves pictures when the depth field is small. Even manual stacking photography would improve photographs of the kind you intend to publish. This does not require a high budget)

There are a few typographical errors in the text and on the cladogram (Italics, space,...). The authors should check that.

(2) Language issue. I have the regret to say that the overall quality of the English is quite poor (except for the descriptions where it is more or less OK). This is the weakest point of this paper (and actually its only weak point). I provide language suggestions for the descriptions only in the line by line comments — but not for the rest of the text, as this is not my job of referee, and as I am not myself an English native speaker. I understand that English is not easy for Chinese authors (I guess that the first author is a young scientist with still little experience; the second author is an experienced scientist), but the next version of the manuscript MUST be checked by an English native speaker with a good knowledge in zoology and marine biology (whether it is re-submitted to PeerJ or to another journal).

(3) Comments line by line.

Line 37–38 "Hippolyte Leach, 1814 are commonly perching in the seaweed of tropical and subtropical oceans"
Not all Hippolyte species are associated with seaweeds. Some are obligatory or facultative symbionts of other organisms. The genus Hippolyte is not only found in tropical/subtropical seas but also in temperate seas. Hippolyte varians has been recorded just above the Arctic circle in Norway (d'udekem d'Acoz 2007).

Line 133 Large-sized shrimp of Hippolyte
I would not say "large-sized". There are much larger species in the genus. For example Hippolyte inermis can reach at least 6 mm CL.

Line 185 "wide, third segment 2.06–2.12 times". use "subsegment" as elsewhere.

Line 2010 "pindling". I am not an English native speaker and that's the first time I see that term. Check with the person who will revise your English that this term is appropriate.

Line 229. "Middle-sized shrimp of Hippolyte". Again, the concepts of large, medium-sized or small are very subjective (I would have said small). Maybe it is better not to refer to size at the beginning of the descriptions.

Line 229. "Outline soft". "Soft" is not the appropriate term. Do you mean robust?

Line 233. "distal teeth in female specimens". "Distal teeth" should become "subdistal tooth".

Line 237: "knoblike" should become "knob-like".

Line 249. I think that "middle-length" should become "mid length".

Line 251. "samll". Do you mean "samll" ?

Line 260. "pudgy palp". I don't think that "pudgy" is the appropriate term.

Line 261 "in low". Do you mean "in low extremity" ?

Line 265. "dactylar segment arch". Do you mean "dactylar segment arched" ?

Line 267. "mid-lenght" should become "mid-length"

Line 271. "mid-lenght" should become "mid-length"

Line 271 "slightly reaching to". Do you mean "just reaching", or "slightly overreaching"?

Line 280. "segment" should become "subsegment".

Lines 280-281 "with tiny setula and long simple setae". I see long distal setae on your drawing, but no "tiny setula". What do you mean by "tiny setula"?

Lines 336-339. " Our specimens of Hippolyte chacei sp. nov. were captured among Sargassum sp. together with Hippolyte cf ventricosa. This may indicate that the two species occupy similar ecological niche, but the ratio of prisal nearly reached to 1:8. We speculate that H. chacei sp. nov. probably remain a predicament in the interspecific competition."

It is quite difficult to explain the difference of frequencies between Hippolyte species in Sargassum beds. Several hypotheses are possible. Here are a few examples. (1) The one proposed by the authors is correct: it is simply rarer. (2) The main habitat of H. chacei is not Sargassum (Sargassum would only be an accessory substrate). (3) H. chacei would have preference for deeper waters and only low densities of specimens would occur at the depths sampled by the authors. (4) H. chacei would prefer environment with a stronger or less strong wave exposure than on the sampling site. (5) Its very high number of teeth on the incisor process suggests a trophic niche distinct from that of other species. this might also play a role in the frequency and micro-distribution of the species.

Line 377. " During the biodiversity surveys". Please be a bit more explicit about these surveys.

Line 377. Hippolyte cf ventricosa. Have you looked at the following characters: number of teeth on mandibular palp; ratio length/width of scaphocerite; number of teeth on fingers of pereiopods 1–2; proportions of the subsegments of pereiopod 2 ? Perhaps will you find differences with the syntypes on these characters. This is just a suggestion.

Lines 403-405 " And in the future, a new taxon established based on integrative datum,eg morphological data, genetic data, and ecological data and so on, will be more valuable and credible."
Personally I would suppress this sentence.

The figures 2 and 11a (colour photographs) are the same. I suggest to suppress the figure 2.

Figure 6. The scale bar is oblique. It should be parallel with the upper and lower margin of the plate.

Reviewer 2 ·

Basic reporting

no comment

Experimental design

no comment

Validity of the findings

no comment

Additional comments

This study comes from an interesting integrative multispecies analysis that combined ecological and morphological traits with genetic data to describe new species from South China sea. The information is original and as far as I know have not been published before. The title is appropriate and concise and the abstract is representative of the content. The text is well organized and clear, most of the interpretations are justified and the discussion and conclusions are supported by the results and the conclusions are correct. However, I found some points that should be checked by the authors before publication.

I´m happy to see new species described based on integrative analysis (morphology and molecules), since this is a new and good science. However, be careful about inferring phylogenetic relationships from a single gene. The 16Smt is very informative, but in some cases its limitations for resolving relationships should be cautiously told, especially for a group that show biological diversity as the Hippolyte members. This is not a really problem, but be cautious. Another aspect that deserves revision are some photos and drawings, whose quality is bad and needs an improvement.

I have made additional suggestions/comments directly in the text designed to improve the manuscript and that should be checked by the authors.

In conclusion, I consider that the present work is suitable for publication after these revisions, which in my opinion do not question the scientific quality of this work, but just the manner to present some data.

Annotated reviews are not available for download in order to protect the identity of reviewers who chose to remain anonymous.

---

## Round 0.2 · Minor Revisions

I recognize the efforts the authors have done providing a new version of the manuscript that seems clearer, especially on the discussion and having added a identification key for the Hippolyte species in the region. However, there are still a number of comments and suggestions regarding the species description, material and methods and discussion section that need some attention from you before the manuscript to be considered in a form to be accepted. I have included the suggestions and the comments on the Word file attached.

Again, I recommend that the new version should be given a thorough revision of the English language by a professional.

---

## Round 0.3 · Minor Revisions

Thank you for all clarifications provided with the rebuttal letter and the accompanying information. In my opinion it is ready to be accepted and I have only one more suggestion that regards the presentation of the information on the Hainan and Xisha Islands surveys (Lines 68 to 71). In my opinion it is better to present the detailed information on the surveys in a table, instead of having all the dates in a long phrase.

---

## Round 0.4 · accepted · Accept

The description of two new species of Hippolyte, using integrative taxonomy (both morphology and molecular analysis), is important to assess the biodiversity for the China Sea and contributes to the overall knowledge on the Hippolytidae. It would be important to provide a correct and complete legend for the tables.

#